# Genetic Characteristics According to Subgroup of Acute Myeloid Leukemia with Myelodysplasia-Related Changes

**DOI:** 10.3390/jcm11092378

**Published:** 2022-04-23

**Authors:** Dain Kang, Jin Jung, Silvia Park, Byung-Sik Cho, Hee-Je Kim, Yeojae Kim, Jong-Mi Lee, Hoon Seok Kim, Ari Ahn, Myungshin Kim, Yonggoo Kim

**Affiliations:** 1Catholic Genetic Laboratory Center, Seoul St. Mary’s Hospital, College of Medicine, The Catholic University of Korea, Seoul 06591, Korea; oskitty@catholic.ac.kr (D.K.); yeojae@catholic.ac.kr (Y.K.); jongmi1226@catholic.ac.kr (J.-M.L.); hskim11@catholic.ac.kr (H.S.K.); 2Department of Biomedicine & Health Sciences, Graduate School, The Catholic University of Korea, Seoul 06591, Korea; 3Department of Laboratory Medicine, College of Medicine, The Catholic University of Korea, Seoul 06591, Korea; bluejin1227@gmail.com (J.J.); 0124ari@gmail.com (A.A.); 4Department of Hematology, Catholic Hematology Hospital, Seoul St. Mary’s Hospital, College of Medicine, The Catholic University of Korea, Seoul 06591, Korea; silvia.park@catholic.ac.kr (S.P.); cbscho@catholic.ac.kr (B.-S.C.); cumckim@catholic.ac.kr (H.-J.K.); 5Leukemia Research Institute, College of Medicine, The Catholic University of Korea, Seoul 06591, Korea

**Keywords:** RNA-based next-generation sequencing, acute myeloid leukemia with myelodysplasia-related changes, previous history, specific cytogenetic, morphological properties

## Abstract

Acute myeloid leukemia with myelodysplasia-related changes (AML-MRC) includes heterogeneous conditions such as previous history and specific cytogenetic and morphological properties. In this study, we analyze genetic aberrations using an RNA-based next-generation sequencing (NGS) panel assay in 45 patients with AML-MRC and detect 4 gene fusions of *KMT2A*-*SEPT9*, *KMT2A*-*ELL*, *NUP98-NSD1*, and *RUNX1*-*USP42* and 81 somatic mutations. Overall, all patients had genetic aberrations comprising of not only cytogenetic changes, but also gene fusions and mutations. We also demonstrated several characteristic genetic mutations according to the AML-MRC subgroup. *TP53* was the most commonly mutated gene (*n* = 11, 24%) and all were found in the AML-MRC subgroup with myelodysplastic syndrome-defining cytogenetic abnormalities (AML-MRC-C) (*p* = 0.002). These patients showed extremely poor overall survival not only in AML-MRC, but also within the AML-MRC-C subgroup. The *ASXL1* (*n* = 9, 20%) and *SRSF2* (*n* = 7, 16%) mutations were associated with the AML-MRC subgroup with >50% dysplasia in at least two lineages (AML-MRC-M) and were frequently co-mutated (55%, 6/11, *p* < 0.001). Both mutations could be used as surrogate markers to diagnose AML-MRC, especially when the assessment of multilineage dysplasia was difficult. *IDH1*/*IDH2* (*n* = 13, 29%) were most commonly mutated in AML-MRC, followed by *CEBPA* (*n* = 5, 11%), *PTPN11* (*n* = 5, 11%), *FLT3* (*n* = 4, 9%), *IDH1* (*n* = 4, 9%), and *RUNX1* (*n* = 4, 9%). These mutations were not limited in any AML-MRC subgroup and could have more significance as a risk factor or susceptibility marker for target therapy in not only AML-MRC, but also other AML categories.

## 1. Introduction

The current diagnosis of acute myeloid leukemia (AML) is largely dependent on genetic aberrations [1]. In the 2016 WHO classification, gene mutations were included in the category of recurrent genetic aberrations such as *NPM1* and double *CEBPA* mutations. Nonetheless, some AML categories are diagnosed based on bone marrow (BM) morphology and other associated findings. One of the aforementioned categories is AML with myelodysplasia-related changes (AML-MRC), which is diagnosed in patients who have previous history or specific cytogenetic or morphological properties. The former two conditions are more obvious than the latter. Studies have questioned the independent predictive value of myelodysplasia in the absence of high-risk cytogenetic abnormalities in AML [2,3] and have defined multilineage dysplasia with more restrictive criteria such as micromegakaryocytes and hypogranulated neutrophils [4]. However, diagnosis remains difficult even for an experienced hematopathologist. In recent years, next-generation sequencing (NGS) has been widely used in clinics and has established the genetic characteristics and their significance in each disease category.

In this study, we analyzed the genetic aberrations in AML-MRC using an RNA-based NGS panel assay and detected gene fusions, mutations, and expressions. We compared the genetic profile among AML-MRC subgroups and endeavored to determine characteristic genetic mutations according to subgroup and to elucidate their clinical significance.

## 2. Materials and Methods

### 2.1. Patients and Samples

We evaluated all consecutive patients who were diagnosed with AML-MRC at Seoul St. Mary’s Hospital, College of Medicine, The Catholic University of Korea, from 2013 to 2018. Patients’ medical records, including history of myelodysplastic syndrome (MDS), myelodysplastic syndrome/myeloproliferative disorder (MDS/MPN), or chemotherapy; laboratory data, BM morphology, and immunophenotypes, were reviewed. Cytogenetic abnormalities were classified according to the 2020 International System for Human Cytogenetic Nomenclature guidelines [5]. Patients were classified into the following three subgroups: (1) patients with history of prior MDS or MDS/MPN (AML-MRC-H), (2) patients with MDS-defining cytogenetic abnormalities (AML-MRC-C), and (3) patients with >50% dysplasia in at least two lineages (AML-MRC-M). The diagnosis and classification were confirmed independently by three hematopathologists. The risk category was determined following the 2017 European LeukemiaNet (ELN) classification [6]. The study received Institutional Review Board approval from Seoul St. Mary’s Hospital, The Catholic University of Korea (IRB No. KC22RISI0078).

### 2.2. Molecular Analysis

Anchored multiplex PCR-based enrichment RNA-sequencing libraries were generated from 250 ng of RNA using the ArcherDx FusionPlex^®^ Myeloid assay for Illumina (ArcherDx, Boulder, CO, USA) according to the manufacturer’s instructions. Briefly, reverse transcription using random primers was performed for synthesis of cDNA, followed by end repair and adenylation steps. Cleanup of cDNA using Agencourt^®^ AMPure^®^ XP beads and ligation of molecular barcode (MBC) adapters and universal primer sites was performed. The MBC adapter-attached cDNA was amplified by the GSP1 primer pool and primer complementary to universal primer site, and a second PCR using the GSP2 primer pool was performed. The libraries were quantitated using a KAPA Universal Library Quantification Kit (Kapa Biosystems, Woburn, MA, USA), normalized, and loaded into NextSeq (Illumina). Data were analyzed by Archer^®^ Analysis version 5.1.7 (ArcherDX). For AMP-based NGS analysis, variants were selected and annotated using analytics algorithms and public databases [7]. The limit of detection for variant calling was set at 2%. Normalized RNA expression values were calculated by dividing the unique RNA reads for each GSP2 by the arithmetic mean of the unique RNA reads for all control GSP2s included in the panel. Relative RNA expression values were reported in RNA_expression_visualization.tsv and were visualized using heat maps. Each heatmap showed samples in columns and binned normalized per GSP2 RNA expression values (0–9) in rows.

*TP53* and *FLT3*-internal tandem duplication mutations were separately analyzed using Sanger sequencing and fragment analysis according to methods used in previous studies, respectively [8,9]. Recurrent gene fusions were analyzed by multiplex reverse transcriptase-PCR (Bio-Rad Laboratories, Hercules, CA, USA) [10].

### 2.3. Statistical Analysis and Response Assessment

Categorical variables were compared using the Chi-square test or Fisher’s exact test, while continuous variables were analyzed with the Mann–Whitney U test and Kruskal–Wallis H test. Overall survival (OS) curves were plotted using the Kaplan–Meier method and were analyzed with the log-rank test. Results were expressed as the hazard ratio with a 95% confidence interval (95% CI). For multivariate analysis, variables with a *p*-value <0.10 in the univariate analysis were entered into a Cox proportional hazards model or proportional hazards model for a subdistribution of competing risk factors. All statistical analyses were performed using SPSS, version 13.0 (SPSS, Inc., Chicago, IL, USA) and R software (version 3.4.1, R Foundation for Statistical Computing, Vienna, Austria, 2017).

## 3. Results

### 3.1. Patient Characteristics

Patient characteristics (total *n* = 45) are summarized in Table 1. The median age at diagnosis was 60 years (range 26–87). Three (7%) patients had a diagnosis of AML-MRC-H. A total of 24 (53%) patients had a diagnosis of AML-MRC-C, including complex karyotype in 17 (71%), monosomy 5 or del(5 q) in 1 (4%), monosomy 7 or del(7q) in 5 (21%), and del(11q) in 1 (4%) patient. Eighteen patients (40%) were diagnosed as AML-MRC-M solely on the basis of dysplasia. The ELN risk based on cytogenetics and presence of *FLT3-ITD*, *NPM1*, *RUNX1*, *ASXL1*, and *TP53* mutations [6] categorized 34 (76%) patients as adverse and 11 (24%) as intermediate. The median age was similar among the AML-MRC subgroups (AML-MRC-M: 57 years (34–81); AML-MRC-C: 60.5 years (26–87); AML-MRC-H: 69 years (61–76)). ELN risk categories were significantly different among AML-MRC subgroups (*p* = 0.002). Most of the AML-MRC-C were included in the adverse group (23/24, 96%), while 56% of AML-MRC-M (10/18) and 33% of AML-MRC-H (1/3) were in the adverse group.

Panmyeloid markers such as CD13, CD33, and cytoplasmic myeloperoxidase were positive in 34 (76%), 43 (96%), and 36 (80%) patients, respectively. Monocytic markers CD11c and CD64 were positive in 32 (71%) and 12 (27%) patients, respectively. The aberrant expression of CD7 and CD56 was observed in 13 (29%) and 12 (27%) patients, respectively.

### 3.2. Genetic Characteristics and Their Association with AML-MRC Subgroups

The genetic landscapes of all patients are shown in Figure 1. We detected a total of 86 genetic aberrations in 39 patients (87%), 4 gene fusions of *KMT2A-SEPT9*, *KMT2A-ELL, NUP98-NSD1*, and *RUNX1-USP42*, and 82 somatic mutations in 20 genes. The median number of somatic mutations was one per patient (interquartile range, IQR: 1–3 mutations), and 44% of patients (*n* = 20) had more than one mutation. When combined with cytogenetics, all AML-MRC cases had at least one genetic aberration. The most frequent mutation was present in *TP53* (*n* = 11, 24%), followed by *ASXL1* (*n* = 9, 20%), *IDH2* (*n* = 9, 20%), *SRSF2* (*n* = 7, 16%), *CEBPA* (*n* = 5, 11%), *PTPN11* (*n* = 5, 11%), *FLT3* (*n* = 5, 11%), *IDH1* (*n* = 4, 9%), and *RUNX1* (*n* = 4, 9%). All *TP53* mutations were observed in AML-MRC-C (*p* = 0.002), specifically in patients with complex karyotype (*p* < 0.001) and monosomy 5 or del(5q) (*p* = 0.001). On the other hand, *ASXL1* and *SRSF2* mutations were more commonly detected in AML-MRC-M compared to AML-MRC-C (*p* = 0.032 and 0.024, respectively) and were frequently co-mutated (55%, 6/11, *p* < 0.001). Patients with *ASXL1* mutation were older, with a median age of 69 years (46–87), compared to those without the mutation (median 58 years, 26–81, *p* = 0.048). They showed a lower expression of *CD274* and *WT1* genes (*p* = 0.020 and 0.041, respectively). Patients with *SRSF2* were older, with a median age of 73 years (56–87, *p* = 0.007), and showed a lower expression of *CD274*, *IRF8*, *MECOM*, and *PDCD1* genes (*p* = 0.026, 0.024, 0.049, and 0.045, respectively) compared to those without the mutation. Patients with the *ASXL1/SRSF2* co-mutation were older, with a median age of 73 years (69–87), compared to those without the co-mutation (*p* = 0.002). *IDH1* and *IDH2* (*n* = 13, 29%) were commonly mutated in AML-MRC and were mutually exclusive. Patients with the *IDH1/IDH2* mutation showed higher proportions of blasts in peripheral blood (52% vs. 10.5%, *p* = 0.006) and BM (81% vs. 37%, *p* < 0.001). In terms of immunophenotypes, the *TP53* mutation was associated with HLA-DR negativity (*p* = 0.012) and the *ASXL1* mutation was associated with CD33 negativity (*p* = 0.004). *CEBPA* and *PTPN11* mutations were associated with cy-MPO negativity (*p* = 0.009 and *p* = 0.009, respectively). *PTPN11* was associated with aberrant CD7 expression (*p* = 0.007) (Appendix A). In addition, the patients with the *IDH1*/*IDH2* mutation showed a higher expression of *ABL1*, *FLT3*, and *RUNX1* (*p* = 0.010, 0.035, and 0.007, respectively) and a lower expression of *IRF8*, *MECOM*, and *MYH11* (*p* = 0.014, 0.017, and 0.033, respectively) compared to those without the mutation (Appendix A).

### 3.3. Clinical Outcomes of Patients with AML-MRC Based on Disease Subgroup and Therapy

Excluding 6 patients (13.3%) without treatment, 31 (68.9%) and 8 patients (17.8%) received intensive chemotherapy and a low-intensity treatment, respectively (Table 1). In detail, among the 31 patients receiving intensive chemotherapy, 12 patients were treated with intensive chemotherapy only, while 19 patients underwent allogeneic stem cell transplantation. Among theeight8 patients receiving the low-intensity treatment, five and three patients were treated with a hypomethylating agent (HMA) and low-dose cytarabine, respectively. There were no significant differences in treatment modalities between the AML-MRC groups. When compared AML-MRC-M with the -C groups, the OS was significantly different; the estimated OS was 13.7 and 5.4 months in the AML-MRC-M and -C groups, (*p* = 0.004), respectively, after the median follow-up period of 84.8 months for survivors (Figure 2a). This significant survival difference was identically observed when the analysis was performed for patients receiving any treatment (*p* = 0.010) or only those who had undergone intensive chemotherapy (*p* = 0.016) (Figure 2b,c). When analyzed within the AML-MRC-C group, the *TP53* mutation predicted a shorter OS not only in all patients (*p* = 0.006), but also in those receiving any treatment (*p* = 0.010) in the univariate analysis for OS (Table 2). In the multivariate analysis for OS, the AML-MRC subgroup had an independent prognostic value after adjusting for age, white blood cell count, and BM blasts at diagnosis. In addition, AML-MRC-C compared to AML-MRC-M showed a significantly worse outcome, with a threefold higher hazard ratio for death (*p* = 0.003) (Table 3).

## 4. Discussion

AML-MRC has been estimated to represent 24–35% of all AML cases and is more commonly seen in older AML patients [11]. In this study, we performed an RNA-based NGS panel assay in AML-MRC and comprehensively analyzed gene fusions, mutations, and expressions. We found four gene fusions, two with *KMT2A* rearrangement. The other two fusions were *NUP98-NSD1* and *RUNX1*-*USP42*, which could be missed on a cytogenetic analysis. *NUP98-NSD1* was reported at a relatively low frequency in AML and MDS and had an impact on poor prognosis [12,13]. The *RUNX1*-*USP42* fusion is a rare *RUNX1* rearrangement in AML and further emphasizes the need for the collection of additional cases [14]. The genetic profile was comparable to results from previous studies (Appendix A) with characteristic genetic mutations according to the AML-MRC subgroup. We also observed that the genetic profile and the prognosis of AML-MRC were not uniform, but significantly differed by the AML-MRC subgroup.

First, the *TP53* mutation was most common in AML-MRC and highly associated with AML-MRC-C. All cases with the *TP53* mutation had complex karyotypes, while a subset of cases with complex karyotype did not have the *TP53* mutation, indicating that complex karyotypic changes generally precede the mutation [15,16]. Both AML-MRC-C and *TP53* mutations were considered adverse prognostic markers, as replicated in this study. Moreover, the *TP53* mutation itself was an adverse prognostic marker within AML-MRC-C. These results collectively indicated that *TP53* mutations are associated with an extremely poor overall survival not only in AML-MRC, but also in the AML-MRC-C subgroup [17].

Second, *ASXL1* and *SRSF2* mutations were associated with AML-MRC-M, and co-mutations were frequent. Results from this and a previous studies showed that *ASXL1*/*SRSF2* co-mutated AML was associated with old age, AML-MRC, and monocytic differentiation [18]. The *ASXL1* mutation was defined as an adverse prognostic marker in AML [6], but we did not confirm a trend towards an inferior outcome in patients with *ASXL1* or the *ASXL1*/*SRSF2* co-mutation. We carefully postulated that *ASXL1* mutated AML consists of heterogeneous cases with morphological signs of dysplasia [19], which exhibit a wide range of prognoses. In addition, the mutation of *SRSF2* as well as *ASXL1* could have a potential role as a surrogate marker in the AML-MRC-M subgroup, especially when the morphological assessment of multilineage dysplasia is difficult [19].

Third, *IDH1* and *IDH2* mutations were detected in all subgroups of AML-MRC without prognostic significance. IDH mutations were also common genetic alterations in AML and MDS. Among AML-MRC, *IDH1* and *IDH2* mutations have been reported at frequencies of approximately 4% and 21%, respectively [20]. A recent study revealed that the *IDH1* mutation was associated with myeloid dysplasia in mice, which exhibited anemia, ineffective erythropoiesis, and increased immature progenitors and erythroblasts [21]. However, there is a considerable number of AMLs with a *IDH1*/*IDH2* mutation categorized into groups other than AML-MRC in cases without definable multilineage dysplasia. The *IDH1*/*IDH2* mutation has been scrutinized more as a susceptible genetic marker for current target therapies [22,23]. Taken together, these findings indicate that the presence of *IDH1*/*IDH2* mutations is informative for therapeutic planning rather than AML categorization [24].

The last group included *CEBPA*, *PTPN11*, *FLT3*, and *RUNX1* mutations. These mutations were detected in all AML-MRC subgroups. Previous studies demonstrated that dysplasia was not relevant for AML with *CEBPA* and *RUNX1* mutations, and RAS pathway mutations and *FLT3*-ITD were significantly more frequent in cases without evaluable erythroid cells [25,26]. Accordingly, we carefully propose that these mutations could have significance as prognostic factors [27], and the classification of patients with these mutations as AML-MRC remains to be clarified. In particular, an understanding of genetic characteristics of each AML-MRC could affect treatment decisions and the therapeutic approach using a novel agent prioritized over conventional chemotherapy [28,29].

For treatment decisions, the diagnosis of AML-MRC by itself may have potential clinical implications. In a previous study by Seymour et al., authors revealed that the use of azacitidine improves clinical outcomes in older patients with AML-MRC as compared to conventional care regimens such as low-dose cytarabine [30]. Further, within [28] a similar diagnosis of AML-MRC, AML-MRC-C showed a worse independent outcome when compared to AML-MRC-M in this present study. These heterogeneities in clinical outcomes suggest that patients who are diagnosed as AML-MRC cannot be regarded as a homogeneous group, but should be considered separately based on the AML-MRC types. In particular, the understanding of each AML-MRC subtype may affect treatment decision, and the therapeutic approach using the novel agent CPX−351 could be prioritized over conventional chemotherapy for medically fit patients with AML-MRC-C and AML-MRC-H. Currently, while the approval of venetoclax in combination with azacitidine did not specifically investigate AML-MRC patients, the incidence of composite complete remission with this combination versus azacitidine alone was notably improved across all AML risk groups, including adverse cytogenetic risk and secondary AML [31], which might suggest potential benefit in the AML-MRC subgroup [32].

Although we clearly demonstrated the characteristic genetic changes in AML-MRC, several limitations were noted. We did not fully evaluate MDS-associated genetic mutations because we used an RNA-based NGS panel assay [33]. We did not include other AML categories whose diagnoses were based on morphology. Issues of total sample size and a comprehensive analysis linked with gene expression need to be addressed in further studies.

In summary, AML-MRC is composed of heterogenous cases with different risk categories and genetic characteristics. Most AML-MRC patients had genetic aberrations, including gene fusions and mutations, as well as cytogenetic changes. In terms of genetic mutations, AML-MRC showed characteristics according to subgroup, and each had its own significance. The *TP53* mutation was closely associated with AML-MRC-C and the extremely poor outcome in AML-MRC. *ASXL1* and *SRSF2* mutations were associated with AML-MRC-M and could be used as surrogate markers to diagnose AML-MRC. Mutations in *IDH1*/*IDH2*, *CEBPA*, *PTPN11*, *FLT3*, and *RUNX1* were not limited to any AML-MRC subgroup and could be more significant as risk factors or susceptible markers for target therapy in not only AML-MRC, but also other AML categories.

## Figures and Tables

**Figure 1 jcm-11-02378-f001:**
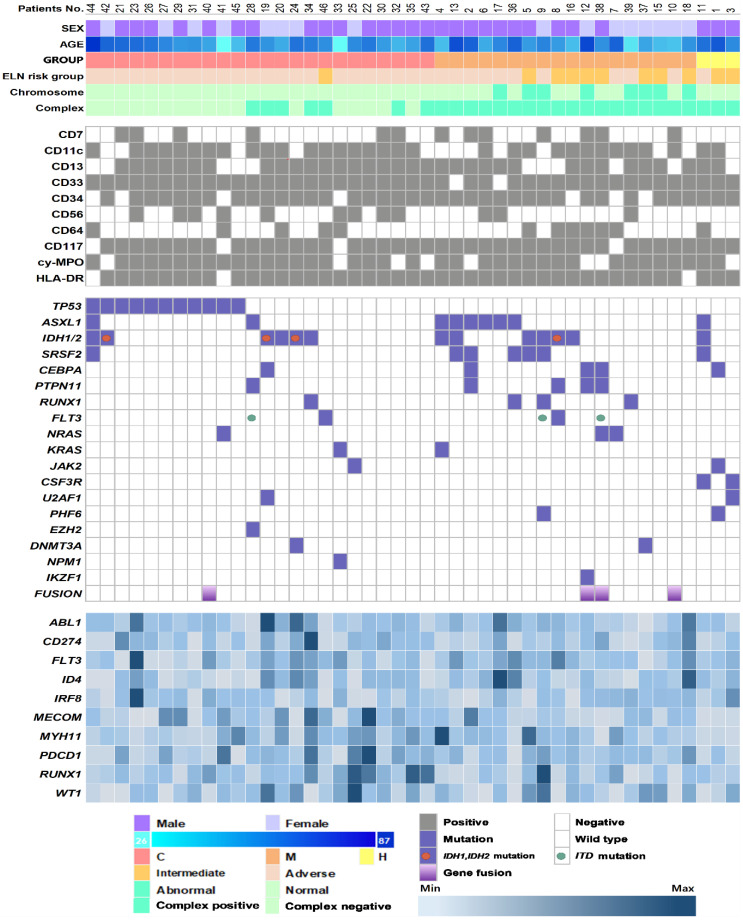
Genetic profile of acute myeloid leukemia with myelodysplasia-related changes (AML-MRC) patients in this study. Data are shown for 45 patients. Each column represents one patient, and each row represents genetic or clinical information. Information on immunophenotypes (gray), mutations (purple), and expressions (blue) is indicated by color and color intensity. Group C: patients with myelodysplastic syndrome (MDS)-defining cytogenetic abnormalities; M: patients with >50% dysplasia in at least two lineages; H: patients with history of prior MDS or MDS/myeloproliferative neoplasm; 2017 ELN risk group was the risk category determined following the 2017 European Leukemia NET.

**Figure 2 jcm-11-02378-f002:**
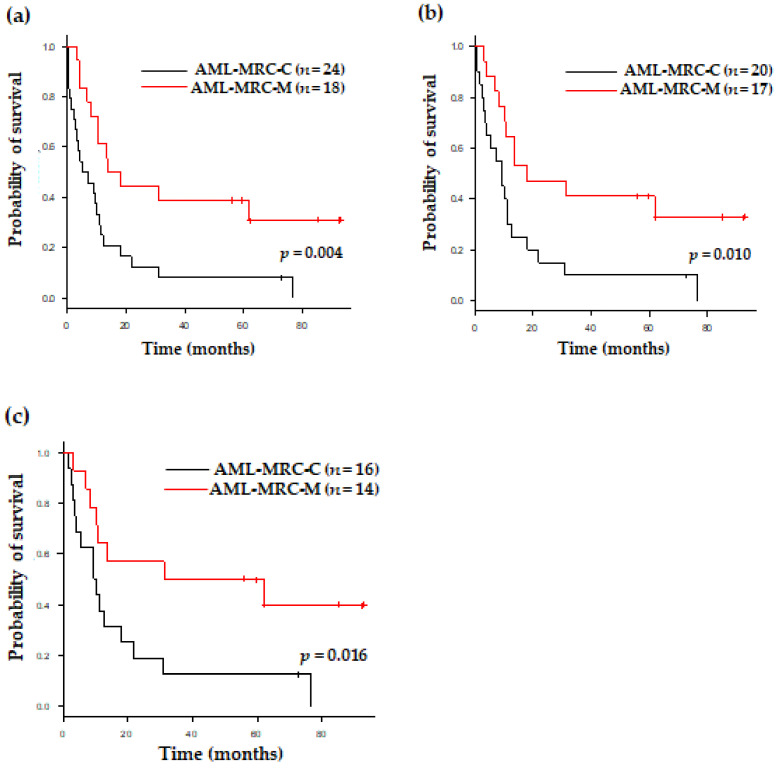
Overall survival of all patients according to AML-MRC subtype (**a**) in all patients, (**b**) among treated patients, and (**c**) among patients who underwent intensive chemotherapy.

**Table 1 jcm-11-02378-t001:** Patient, disease, and treatment characteristics according to acute myeloid leukemia with myelodysplasia-related changes (AML-MRC) subgroup.

Variables	Overall	AML-MRC-H	AML-MRC-C	AML-MRC-M	*p*-Value
Age at diagnosis, years					0.324
Median (range)	60 (26–87)	69 (61–76)	60.5 (26–87)	57 (34–81)	
Sex, *n* (%)					0.312
Male	29 (64.4)	3(100)	16 (66.7)	10 (55.6)	
Female	16 (35.6)	0	8 (33.3)	8 (44.4)	
WBC count at diagnosis					0.780
Median (range)	3.19 (0.78–363.5)	12.64 (0.9–42.85)	3.67 (0.78–169.6)	2.91 (0.84–363.5)	
WBC group at diagnosis, *n* (%)					0.629
<50 × 10⁹/L	41 (91.1)	3 (100)	21 (87.5)	17 (94.4)	
≥50 × 10⁹/L	4 (8.9)	0	3 (12.5)	1(5.6)	
2017 ELN risk group, *n* (%)					0.002
Favorable	0	0	0	0	
Intermediate	11 (24.4)	2 (66.7)	1 (4.2)	8 (55.6)	
Adverse	34 (75.6)	1 (33.3)	23 (95.8)	10 (44.4)	
Treatment, *n* (%)					0.469
Low-intensity treatment	8 (20.5%)	3 (17.6%)	4 (20.0%)	1 (50.0%)	
Intensive chemotherapy	31 (79.5%)	14 (82.4%)	16 (80.0%)	1 (50.0%)	
Treatment detail, *n* (%)					1.000
Hypomethylating agent (HMA)	5 (12.8%)	3 (17.6%)	1 (5.0%)	1 (50.0%)	
Low-dose cytarabine	3 (7.7%)	0 (0.0%)	3 (15.0%)	0 (0.0%)	
Intensive chemo only	12 (30.8%)	4 (23.5%)	8 (40.0%)	0 (0.0%)	
Intensive chemo + transplantation	19 (48.7%)	10 (58.8%)	8 (40.0%)	1 (50.0%)	

AML-MRC-H: patients with history of prior myelodysplastic syndrome (MDS) or MDS/myeoloproliferative nepoplams; AML-MRC-C: patients with MDS-defining cytogenetic abnormalities; AML-MRC-M: patients with >50% dysplasia in at least two lineages; WBC: White Blood Cell; 2017 ELN risk group was the risk category determined following the 2017 European LeukemiaNET.

**Table 2 jcm-11-02378-t002:** Overall treatment outcomes: univariate analysis.

Variables	Hazard Ratio	95% CI	*p*-Value
Age	1.049	1.016–1.084	0.004
WBC at diagnosis	1.000	1.000–1.000	0.273
BM blast	1.007	0.993–1.020	0.318
AML-MRC subgroup			0.018
Mutation			
*TP53*	4.580	2.156–9.729	<0.001
*ASXL1*	0.844	0.371–1.921	0.686
*IDH2*	0.687	0.286–1.649	0.100
*SRSF2*	1.025	0.428–2.458	0.955
*CEBPA*	1.281	0.494–3.323	0.611
*PTPN11*	1.250	0.482–3.243	0.647
*IDH1*	2.139	0.734–6.235	0.164
*RUNX1*	1.160	0.408–3.295	0.781

CI: confidence interval; WBC: while blood cell count; BM: bone marrow; AML-MRC: acute myeloid leukemia with myelodysplasia-related changes.

**Table 3 jcm-11-02378-t003:** Multivariate analysis for overall survival.

Characteristics	*p*-Value	Hazard Ratio (95% CI)
Age (continuous variable)	0.003	1.061 (1.021–1.103)
WBC at diagnosis (continuous variable)	0.451	1.000 (1.000–1.000)
BM blasts (continuous variable)	0.655	1.003 (0.989–1.017)
AML-MRC-C vs. AML-MRC-M	0.004	3.055 (1.425–6.547)

CI: confidence interval; WBC: while blood cell count; BM: bone marrow; AML-MRC: acute myeloid leukemia with myelodysplasia-related changes; AML-MRC-C: patients with MDS-defining cytogenetic abnormalities; AML-MRC-M: patients with >50% dysplasia in at least two lineages.

## Data Availability

The data presented in this study are available on request from the corresponding author. The data are not publicly available due to ethical concern.

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
