# Peer review of "Genetic Characteristics According to Subgroup of Acute Myeloid Leukemia with Myelodysplasia-Related Changes"

_jcm, 2022, doi:10.3390/jcm11092378_

Round 1

Reviewer 1 Report

The paper «Genetic characteristics according to subgroup of acute myeloid leukemia with myelodysplasia-related changes» by Kang and coworkers is submitted to Journal of Clinical Medicine. In the paper they describe acute myeloid leukemia with myelodysplasia-related changes (AML-MRC), in a retrospective study.  They analyzed genetic aberrations using an RNA-based next-generation sequencing (NGS) panel assay in 45 patients with AML-MRC, they demonstrated several characteristic genetic mutations according to AML-MRC subgroup, including TP53, ASXL1, RUNX1 and SRSF2. They divided the AML-MRC group in three; AML-MRC-H (n=3) AML-MRC-C (n=21) and AML-MRC-M (n=17). They found that the AML-MRC-C had the worst outcome.

In general, I think the paper bring “little new to the table”. That AML-MRC patients have dismal prognosis and are associated with high-risk mutation are described in larger studied before. I also have some concerns regarding the classification, as I would have expected some degree of overlaps between the groups; AML-MRC-H, AML-MRC-C, AML-MRC-M groups, as nicely demonstrated in this paper: “Azacitidine improves clinical outcomes in older patients with acute myeloid leukemia with myelodysplasia-related changes compared with conventional care regimens” By Seymour et al in BMC Cancer 2017.

Furthermore, the AML-MRC-H only consist of three patients, and I think this is too little to make any significant conclusions. Hence, I do not think the conclusions are justified by the results in the paper.  

Author Response

Thank you.

Reviewer 2 Report

The authors have performed a single-center retrospective analysis aiming to evaluate the genetic characteristics according to subgroup of acute myeloid leukemia with myelodysplasia-related changes.

The paper is written in a good English, and it can be clear for readers.

The paper is complete, well organized, and can be really interesting for readers.

Discussion could be implemented, better focusing on similar data from literature, and adding a table with data from literature about this setting of patients could be useful for readers.

The idea is good and, even if the number of patients is low, data are interesting and well presented and the paper could be really interesting for readers.

Author Response

Thank you.

Reviewer 3 Report

jcm-1597592

Genetic characteristics according to subgroup of acute myeloid leukemia with myelodysplasia-related changes

The article "Genetic characteristics according to subgroup of acute myeloid leukemia with myelodysplasia-related changes" (jcm-1597592) by Kang D, et al. demonstrated that subtype of AML-MRC and several gene mutations in the patients with AML-MRC using NGS predicted overall survival in the real-world data. These clinical significances are very interesting and informative. On the other hand, I considered that it was preferred to add several analyses and discussion about treatment. Therefore, there are several major and minor issues to be addressed as below.

Major issues

  1. The author should revise the data about treatment more in detail. For example, the number of patients received allogeneic transplantation, treated with hypomethylating agents, and FLT3 inhibitors, etc.

  1. The author should analyze the correlation between age and gene mutations.

  1. The author should add the previous article of targeting agents for AML, such as HMA, gemtuzumab ozogamicin, CPX-351, FLT3-inhibitor, IDH inhibitor, venetoclax in the discussion part.

Minor issues

  1. The author could add the association between gene mutation and phenotype.

  1. The title of table 1 was not correct.

  1. The paragraph of “3.3. Clinical outcomes of patients with AML-MRC based on disease subgroup and therapy ” was described twice.

Author Response

Thank you.

Round 2

Reviewer 1 Report

Although the manuscript is improved, I still find some part of concern.

Of notice, AML-MRC-H patient (only 3) should be left out of graphical presentation. Survival curve with one (!?) patient do not make much scence.

Author Response

Thank you.

Reviewer 3 Report

I considered that this revised article was suitable for acceptation in "JCM".

Author Response

Thank you.
